# Effects of Process Parameters on Microstructure and High-Temperature Oxidation Resistance of Laser-Clad IN718 Coating on Cr5Mo Steel

Zelin Xu [1], Fengtao Wang [1,*], Shitong Peng [1], Weiwei Liu [2] and Jianan Guo [1]

1 Department of Mechanical Engineering, College of Engineering, Shantou University, Shantou 515063, China
2 School of Mechanical Engineering, Dalian University of Technology, Dalian 116024, China
* Correspondence: ftwang@stu.edu.cn; Tel.: +86-0754-86502941

**Abstract:** Cr5Mo steel with high thermal strength is frequently applied as the material for hydrocracking furnace tubes. Nonetheless, Cr5Mo tubes are prone to material failure in a high-temperature environment, threatening production safety. Considering that the IN718 nickel-base superalloy has favorable high-temperature oxidation resistance, the IN718 coating was fabricated on Cr5Mo substrate through laser cladding. The effect of process parameters on the high-temperature oxidation resistance of laser cladding IN718 coating was investigated. The results confirm that laser power and scanning speed affected the eutectic quantity precipitation of this layer, and the eutectic quantity precipitation was positively correlated with the mass gain of the coating. The high-temperature behavior of the coating could be divided into surface oxidation, intergranular corrosion, and material shedding. The scanning speed has a more significant impact on the high-temperature oxidation resistance. When the scanning speed is 15 mm/s, cracks originating in the heat-affected zone could exert a negative impact on the high-temperature oxidation resistance.

**Keywords:** laser cladding; Cr5Mo; IN718; HAZ crack; delamination crack; high-temperature oxidation resistance

## 1. Introduction

Cr5Mo tubes are a part of the furnace that directly heats the gas or liquid flowing in the tube by flame [1,2]. In general, the petrochemical furnace works continuously under high temperatures, high pressure, and corrosive mediums. The heated gas or liquid in the tube is commonly a flammable and explosive hydrocarbon [3]. Overheating and flame licking could damage the furnace tube, which adversely affects production safety and efficiency. Therefore, the high-temperature oxidation resistance of furnace tubes is an essential property for the safe as well as efficient service of the tubular furnace.

Laser cladding is one of the promising methods for manufacturing and repairing different industrial components such as pump shafts and turbine blades [4–7]. IN718 nickel-base superalloy has high strength, satisfying toughness, and corrosion resistance below 650 °C [8,9]. Thus, laser cladding IN718 coating is able to contribute to the improvement of the service life of Cr5Mo substrate in a high-temperature corrosion environment. Xie et al. [10] demonstrated that the increment of the cooling rate could inhibit undercooling, reduce the secondary dendrite arm spacing, and refine the Laves phase in the coating. Zhang et al. [11] prepared IN718 coating on the deformed IN718 alloy plate and investigated the effects of distinct laser cladding speeds on the microstructure and Nb segregation of the coating. These studies indicated that the process parameters vary in the microstructure of the coating, which may further affect the high-temperature oxidation resistance of the IN718 coating. Jia et al. [12] studied the high-temperature oxidation behavior and mechanism of IN718 components after selective laser melting (SLM) in the near-surface area. The results illustrated that increasing the laser energy density could facilitate the high-temperature

oxidation resistance of the IN718 components. However, the laser cladding process differs from the SLM process in being exposed to the atmosphere, and the local temperature difference leads to shrinkage, residual stress, and deformation. Therefore, the effects of process parameters on coating quality and high-temperature oxidation resistance are worth investigating. Under specific process conditions, despite the fact that the surface quality of the laser cladding coating is intact, defects such as cracks and pores potentially occur inside [13]. Mazzarisi et al. mentioned [14] that a high number of cracks was found in areas with higher temperature and thermal gradient variations. Such defects have negative impact on the performance and behavior of AM-produced alloys [15]. In addition to the high-temperature oxidation behavior of the top area of the aforementioned laser cladding coating in the existing research [16], Thouless [17] reported that cracks and delamination through the thickness of the coating would accordingly damage the protective coating. It should be pointed out that there is an absence of research on the effects of process parameter adjustment and defects on the high-temperature oxidation resistance of laser cladding IN718 coating.

This work will study the microstructural evolution and defect generation mechanism of coating under different process parameters, analyze the correlation between process parameters and high-temperature oxidation resistance, and supplement the high-temperature behavior of laser cladding IN718 coating in a high-temperature oxidation environment. Investigating the high-temperature oxidation resistance of laser cladding IN718 coating is beneficial for improving the high-temperature oxidation resistance and service life of IN718 coating. Lastly, an engineering application example of remanufactured Cr5Mo hydrocracking furnace tubes is given.

## 2. Materials and Methods

### 2.1. Substrate and Laser Cladding Powder

This study adopted the Cr5Mo tube as substrate with an external diameter of 180 mm, a thickness of 15 mm, and a length of 250 mm and IN718 powder with a mesh size of 45–100 μm as the laser cladding material. The chemical compositions of these materials are listed in Table 1. The morphology of IN718 powder was nearly spherical, as shown in Figure 1a. The particle size was measured using a DynaPro NanoStar Dynamic Light Scattering (DLS) instrument (Wyatt Technology Corporation, Santa Barbara, CA, USA). The particle-size distribution of IN718 powder is shown in Figure 1b.

**Table 1.** Chemical composition of Cr5Mo substrate and IN718 powder.

| Substrate | Cr | Mo | Mn | C | Si | Nb | Ni | Fe |
|---|---|---|---|---|---|---|---|---|
| Cr5Mo | 5.02 | 0.54 | 0.35 | 0.10 | 0.54 | 0.02 | - | Bal. |
| IN718 | 21.0 | 3.3 | 0.35 | 0.08 | 0.35 | 5.50 | Bal. | 14.50 |

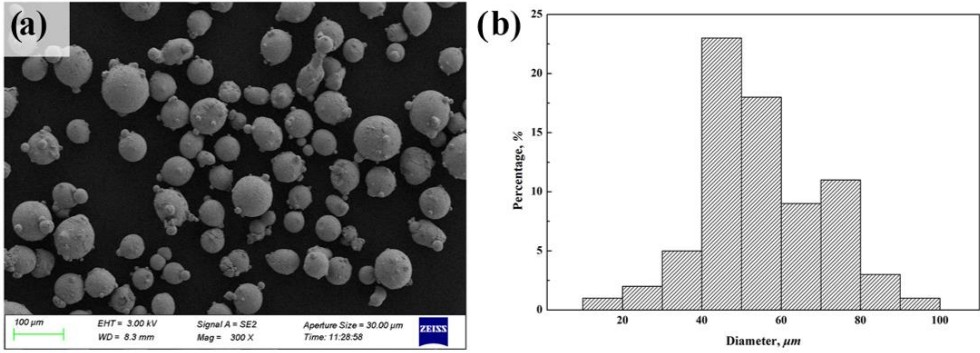

**Figure 1.** IN718 powder characterization: (**a**) SEM image of nearly spherical powder and (**b**) particle-size distribution.

### 2.2. Preparation of Laser Cladding Layer and Test Sample

A TRUMPF 3600 laser machine (spot size of 4 mm) equipped with coaxial powder feeding was used for laser cladding. The parameters of the coating preparation are listed in Table 2. Two critical operational parameters, namely, the laser power and scanning speed, were varied to study their effects on the physical properties of laser cladding coating. Other parameters include an overlap rate of 50% and a powder feeding rate of 15 g/min. As sketched in Figure 2, a total of five groups of the IN718 coatings with different parameters were prepared on Cr5Mo tubes, with average coating thickness of 3 mm and coating width of 15 mm. The distance between the coatings was 20 mm. Samples for testing were prepared from the deposited layer by wire electrical discharge machining (WEDM) (Makino, Tokyo, Japan), which were equipped with a size of 10 mm × 10 mm × 10 mm. A TASi TA603 thermometer (TASI, Suzhou, China) was used to measure the temperature of the substrate about 10 mm from the laser-clad track. The temperature of measured positions after laser cladding was mostly lower than 150 °C.

**Table 2.** Parameters for studying the effect of the laser cladding coating.

| Experimental Group | Laser Power (*W*) | Scanning Speed (mm/s) |
|:---:|:---:|:---:|
| 1600-5 | 1600 | 5 |
| 1600-10 | 1600 | 10 |
| 1600-15 | 1600 | 15 |
| 2000-10 | 2000 | 10 |
| 2400-10 | 2400 | 10 |

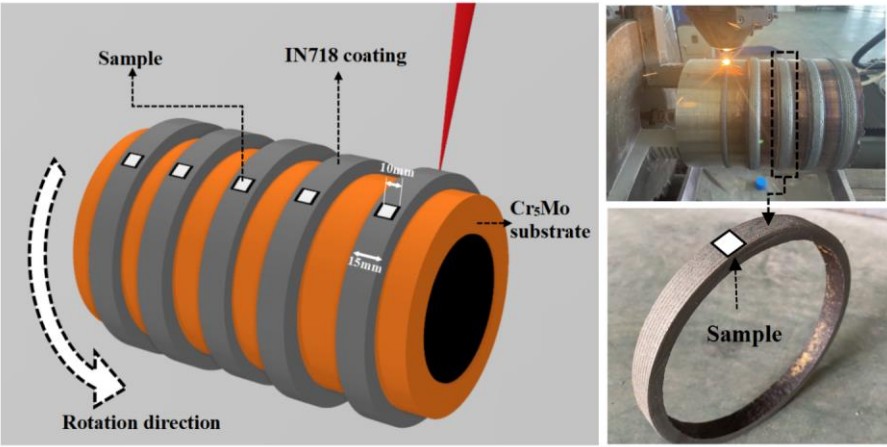

**Figure 2.** Preparation of laser cladding samples with different parameters on Cr5Mo.

### 2.3. Analysis and Testing

For microstructural observation, samples were initially polished and then etched with Marble's reagent (10 g $CuSO_4$ + 50 mL HCl + 50 mL $H_2O$). Microstructure and microhardness were analyzed using an MR5000 metallography microscope and a TMVP-1 microhardness tester. The load for the hardness test was set to 0.5 kg, and the dwell time was set to 10 s. Zeiss Sigma 300 scanning electron microscope (SEM) (Carl Zeiss AG, Oberkochen, Germany) was equipped with energy dispersive X-ray spectroscopy (EDS) for microstructural and chemical analysis. This paper adopted a Panalytical X'Pert'3 powder X-ray diffraction system (XRD) (PANalytical, Almelo, Netherlands) to observe the crystalline structure. Samples were heated in an SX2-5-12A muffle furnace. The increased oxidation weight (mass gain) tested by an EX1035 electronic analytical balance was employed to quantify the high-temperature oxidation resistance [18]. The oxidation resistance of IN718 coatings was quantified at high temperature. The samples were heated at 900 °C for 100 h. Moreover, Minitab software (Minitab 2021) was used for data fitting.

## 3. Results

### 3.1. Microstructure

Figure 3 indicates the XRD spectra of IN718 coating with different process parameters. The phases of the IN718-cladded layer were mainly $\gamma$-Ni (PDF-# 65-0380), $\gamma'$-Ni$_3$(Al, Ti) (PDF-# 18-0872), and $\gamma''$-Ni$_3$Nb (PDF-# 15-0101), which were consistent with the research of Sumit et al. [19] without the appearance of new phases when changing the process parameters. However, the diffraction peak intensity of the samples did change. When the power was 1600 W and the scanning speed was 5 mm/s (sample 1600-5), the peak intensities of $\gamma$-Ni and $\gamma'$-Ni$_3$(Al, Ti) were higher than that of the samples 1600-10 and 1600-15, indicating the more significant proportion of $\gamma$-Ni and $\gamma'$-Ni$_3$(Al, Ti). The decrease in scanning speed reduced the cooling rate, which was conducive to the growth of austenite dendrites. When the scanning speed was 10 mm / s and the laser power increased from 1600 W to 2400 W (samples 1600-10, 2000-10, and 2400-10), the intensity of the second peak significantly increased. As can be seen from the results, the higher laser power can promote the growth of dendrites.

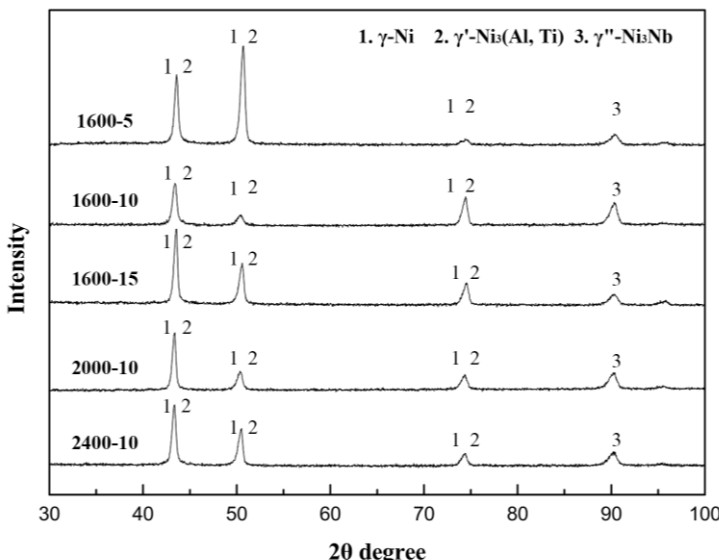

**Figure 3.** XRD spectra of IN718 coating under different process parameters.

Figure 4 displays the effects of power and scanning speed on the primary dendrite arm spacing (PDAS) of five samples. The dendrites among all the samples grew opposite to the heat transfer direction. The refinement degree of the microstructure was compared by measuring the distance between primary dendrites. With an increment in the laser power from 1600 W to 2400 W, the PDAS of IN718 laser cladding layer increased from 19.47 μm to 27.83 μm. Nevertheless, the PDAS decreased from 25.23 μm to 7.16 μm when the scanning speed increased from 5 mm/s to 15 mm/s. These results were in line with prior studies on the influence of laser power and scanning speed on the microstructure [20]. The size of the martensite and lower bainite increased during the process of low alloy steel fabrication through laser melting deposition. With the increase in laser power, the tempered martensite gradually coarsens. In the study of laser selective melting of 316 L stainless steel, the increase in scanning speed could induce grain refinement [21]. In terms of IN718 laser cladding coating, the variation of heat input and cooling rate led to the difference of PDAS.

Figure 5 presents the microstructural evolution of the top region of the laser cladding IN718 coating under different process parameters. It can be seen from Figure 5a–c that when the scanning speed increased from 5 mm/s to 15 mm/s, the top region's dendrites refined, and eutectic quantity increased. As shown in Figure 5b,d,e, when the laser power increased from 1600 W to 2400 W, the dendrite in the top region coarsened, and the eutectic quantity decreased. Figure 6 shows the microstructure evolution of the bottom region of

laser cladding IN718 coating under different process parameters. Du et al. [22] mentioned that columnar dendrites were generated in the bottom region, and equiaxed dendrites were formed in the top region. It can explain the difference of microstructure morphology between Figures 5 and 6.

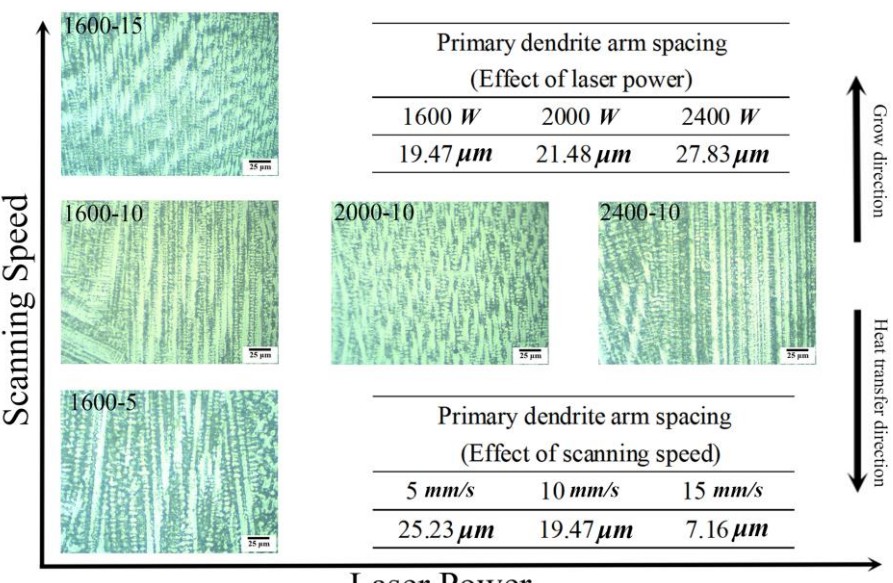

**Figure 4.** Effects of laser power and scanning speed on sample PDAS.

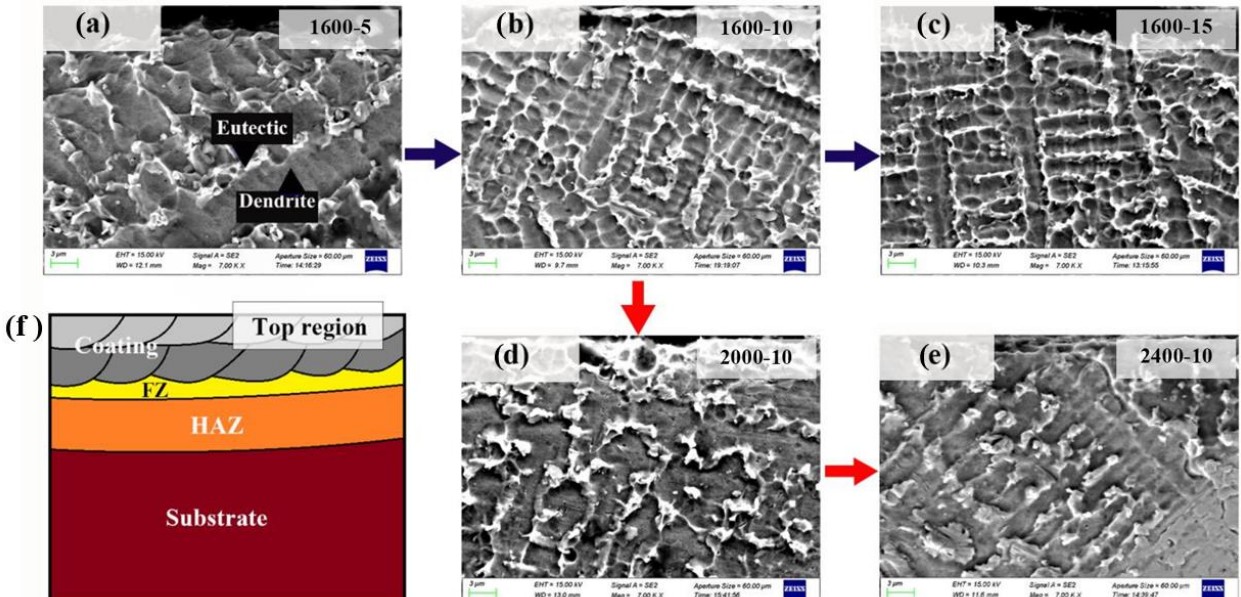

**Figure 5.** Microstructural evolution in the top region of laser cladding IN718 coating under different process parameters (**f**): Scanning speed increased from 5 mm/s to 15 mm/s (blue route (**a**–**c**), and laser power increased from 1600 W to 2400 W (red routes (**b**,**d**,**e**).

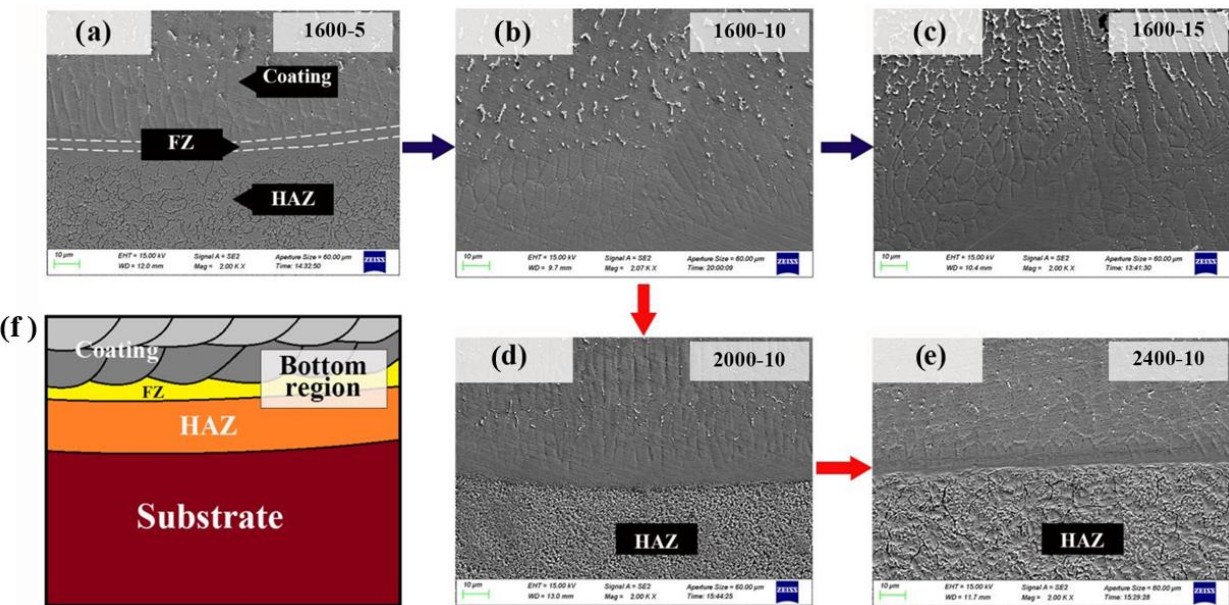

**Figure 6.** Microstructural evolution in the bottom region of laser cladding IN718 coating under different process parameters (**f**): Scanning speed increased from 5 mm/s to 15 mm/s (blue route (**a–c**), and laser power increased from 1600 W to 2400 W (red routes (**b,d,e**)).

In Figure 6a, the bottom region of the coating, the fusion zone (FZ), and the heat-affected zone (HAZ) were observed. As indicated in Figure 6a–c, there was an increased eutectic quantity in the top region when the scanning speed increased from 5 mm/s to 15 mm/s. In Figure 6b,d,e, the eutectic quantity decreased gradually when the laser power increased from 1600 W to 2400 W. Higher cooling rates could result in increased precipitation of eutectic elements. Lower power and faster scanning speed improve the cooling rate. Additionally, the higher cooling rates could eventually increase eutectic compounds while decreasing austenite dendrites. Such phenomena have also been observed in previous work [23].

### 3.2. Hardness Testing

Microhardness testing was measured from the coating surface to the substrate at room temperature, with seven testing locations in a straight line, whose results are listed in Table 3. Figure 7 indicates that the laser power and scanning speed affect the microhardness from the coatings to the substrate. In Figure 7a, the scanning speed (invariant) was 10 mm/s, while the laser power (variable) was increased from 1600 W to 2400 W. In Figure 7b, the laser power (invariant) was 1600 W, while the scanning speed (variable) was increased from 5 mm/s to 15 mm/s. The hardness of the coating and HAZ in Figure 7 are the average values at different positions listed in Table 3.

**Table 3.** Microhardness (room temperature) from coating to the substrate, $HV_{0.2}$.

| - | 1600-5 | 1600-10 | 1600-15 | 2000-10 | 2400-10 |
|---|---|---|---|---|---|
| Coating | 300.8 | 344.3 | 345.6 | 279.8 | 306.4 |
|  | 249.8 | 276.5 | 367.9 | 238.7 | 251.5 |
| FZ | 218.3 | 225.7 | 304.2 | 186.5 | 162.4 |
| HAZ | 284.9 | 324.1 | 400.9 | 335.4 | 325.2 |
|  | 307.3 | 340.8 | 370.8 | 441.4 | 333.8 |
|  | 269.5 | 281.3 | 314.2 | 291.8 | 283.8 |
| Substrate | Average 252.9 | | | | |

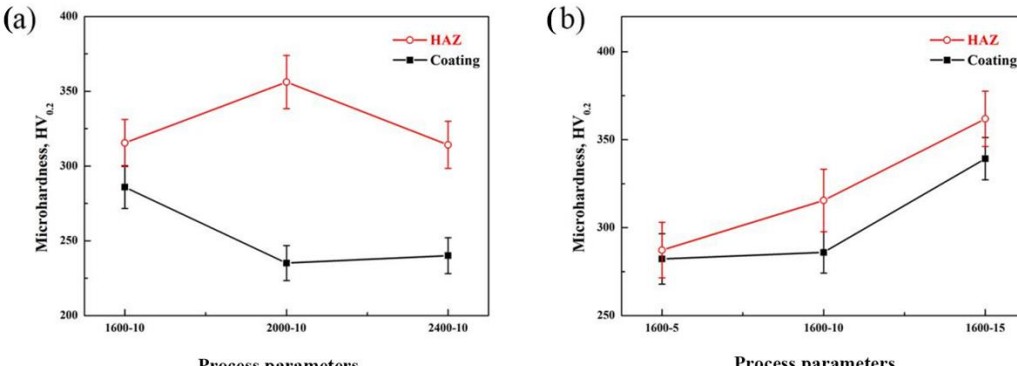

**Figure 7.** Factors affecting the microhardness of coating-substrate: (**a**) effects of laser power and (**b**) effects of scanning speed.

As shown in Figure 7a, with the increase in laser power, the average hardness of the coating decreases from 310.4 $HV_{0.2}$ to 278.9 $HV_{0.2}$, and the hardness of HAZ firstly shows an initial increment and then decreases. When the scanning speed increased (Figure 7b), the average hardness of the coating increased from 275.3 $HV_{0.2}$ to 356.7 $HV_{0.2}$, and the average hardness of HAZ increased from 287.2 $HV_{0.2}$ to 361.9 $HV_{0.2}$. Mankins et al. [24] stated that the primary strengthening mode of nickel-based superalloy stemmed from the precipitation strengthening of the $M_{23}C_6$ phase. Therefore, the change of laser process parameters affected the number of strengthening phases precipitated in the parent phase and subsequently changed the hardness of the coating.

Notably, the hardness of HAZ was generally higher than that of the coating and substrate. Adomako et al. [25] reported that the hardness of IN718 coating decreased continuously from the surface to the FZ due to the depletion of strengthening elements (Nb, Mo, and Ti) in the microstructure, which was also consistent with the present study. Under the condition of laser heating and rapid cooling, martensite was generated in the HAZ region [26]. The martensite had high hardness in the HAZ region.

It can be seen in Figure 7a that the hardness of HAZ first increased and afterwards decreased with the increase in laser power from 1600 W to 2400 W. As shown in Figure 6d,e, the HAZ microstructures of samples with different powers exhibit apparent differences. The SEM images of HAZ in samples with laser powers of 2000 W and 2400 W are observed in Figure 8. With the increase in laser power, the temperature of laser cladding track rose, and the cooling rate decreased, resulting in ferrite recrystallization and grain coarsening (Figure 8b). Microhardness reduction could also be proved when laser power increased from 2000 W to 2400 W.

The HAZ of these two samples was analyzed by EDS elemental mapping. In Figure 8c,d, the EDS elemental mapping results of HAZ in both samples with laser power of 2000 W and 2400 W showed a stable composition. Thus, the possibility of interference from other elements was excluded. It has been proved that changing the power produces two opposite factors in HAZ: With the increase in power, carbide precipitation can increase the hardness of HAZ; grains coarsen, and the hardness of HAZ decreases. Before discussing the high-temperature oxidation resistance of IN718 coating, it is necessary to study the microstructural evolution in HAZ, which determines the service life of IN718 coating.

*3.3. Defects*

Figure 9 provides various types of defects in the IN718 laser cladding coating. It is evident from Figure 9a that a crack starts from the bottom region of the coating and grows towards the surface of the coating. The crack grows along the boundary of dendrites. Combined with the analysis in Figure 6, more eutectic elements, including Laves phases and carbide, were generated in the bottom region of the coating under the condition of low laser power. Carbide and Laves phases promoted the segregation of impurity elements to grain boundaries, leading to low melting compounds [27], which induced

intergranular microfissures due to grain boundary liquation. The increase in HAZ hardness was attributed to the precipitation hardening of the martensite, while it brought HAZ a higher brittleness and a higher crack sensitivity. A crack starting from HAZ can be seen in Figure 9b, which then extended into the coating. In Sample 1600-15, a thermal fatigue crack in HAZ initiated under thermal cyclic loading. The crack started from the HAZ also proved the necessity of studying the microstructural evolution of the coating–substrate bonding region. As shown in Figure 9c, a pore was observed in Sample 1600-15. Reactive and non-reactive gases exist in the process of laser cladding. In Sample 1600-15, the coating did not release the gas promptly due to the rapid solidification at high scanning speed. Therefore, the unreleased gas caused pore defects in the samples.

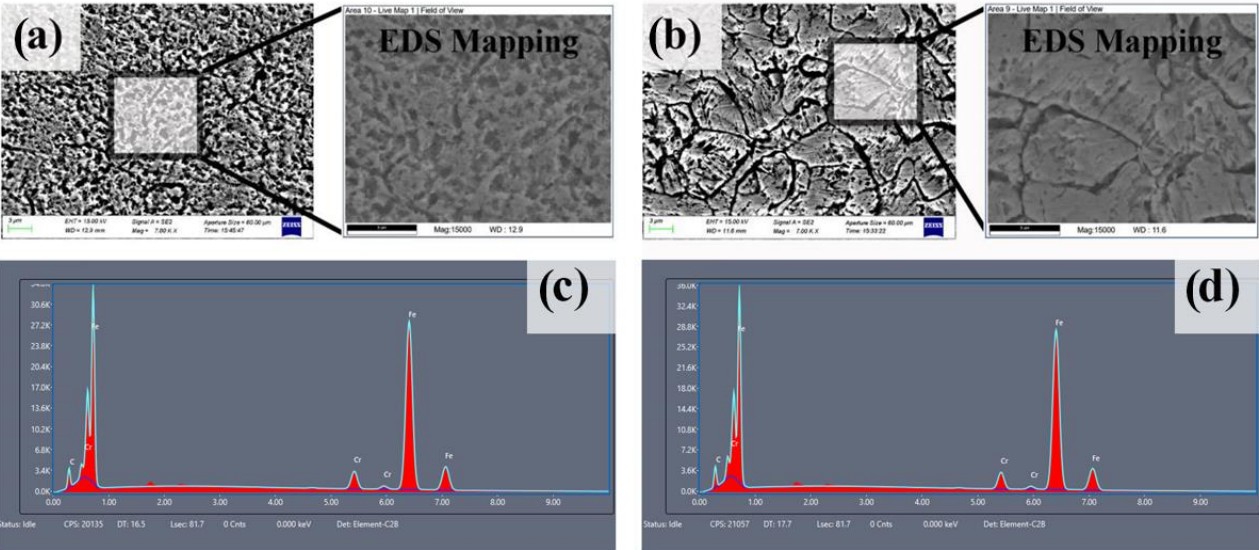

**Figure 8.** SEM images and EDS mapping results of HAZ: samples with laser power of 2000 W (**a**,**c**) and samples with laser power of 2400 W (**b**,**d**).

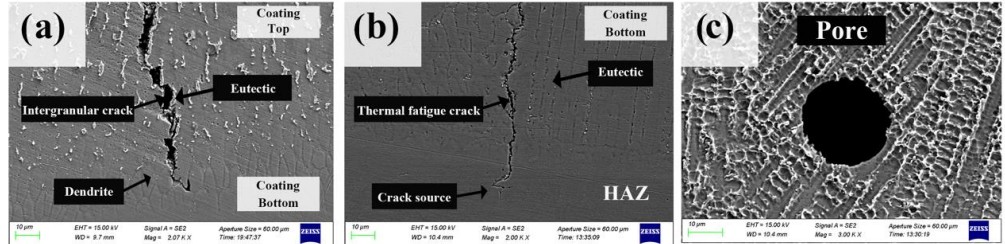

**Figure 9.** Different types of defects in IN718 coating: (**a**) intergranular crack in Sample 1600-10; (**b**) thermal fatigue crack in HAZ of Sample 1600-15; (**c**) pore in Sample 1600-15.

### 3.4. High-Temperature Oxidation Resistance

Nickel-based materials could be oxidized at a high temperature of 900 °C, and an oxide film was formed on the coating surface [28,29]. Figure 10 is the EDS mapping results of the Sample 1600-10 after the high-temperature oxidation treatment at 900 °C. The enrichment of element O in region (I) of the coating indicates that the final oxidation products are some oxides of element Ni, such as $NiFe_2O_4$, $NiO$, and $NiCr_2O_4$. These phases have also been observed by Wu et al. [30]. After high-temperature oxidation treatment, the eutectic products in region (II) below the oxide film were significantly reduced. The content of Nb in region (III) increased with the increase in the eutectic content. The formation of oxide was closely related to the distribution of the Ni3Nb phase in the top region of the coating. Element Nb tended to be enriched below the oxide film of IN718 after high-temperature treatment, which was attributed to the eutectic generation near the surface and the movement of the element Nb.

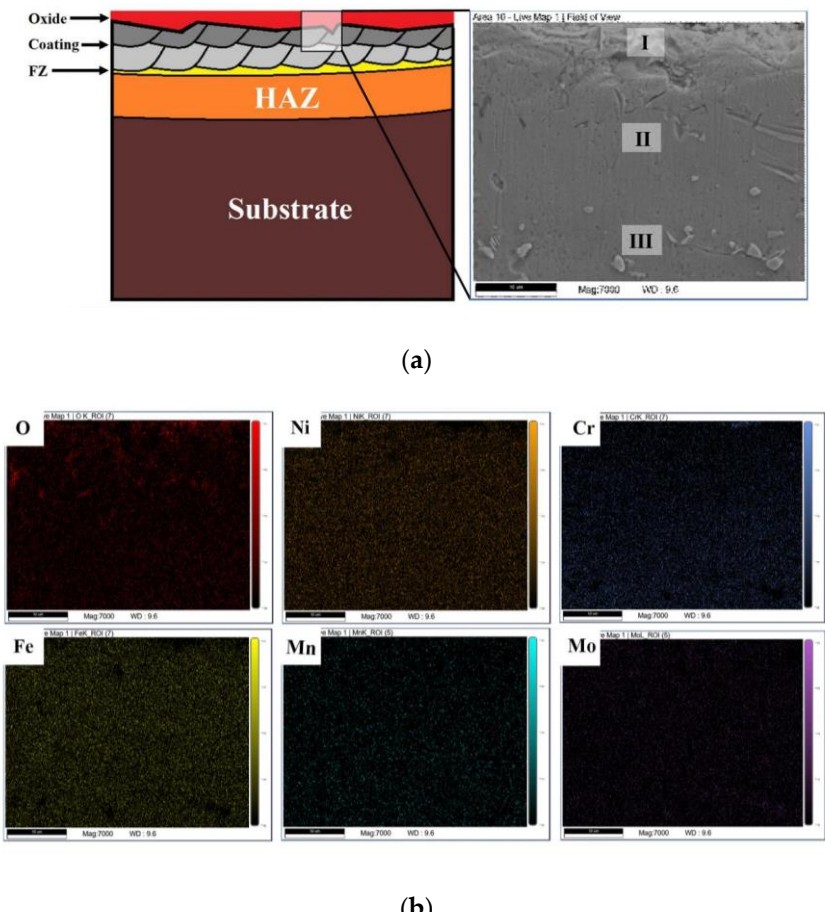

(a)

(b)

**Figure 10.** SEM (**a**) and EDS mapping result (**b**) of Sample 1600-10 after high-temperature oxidation treatment.

In Figure 11a, the enriched eutectic particles led to the cracking of the Sample 1600-10. As stated by Prakash et al. [31], the element Nb and the distribution of second-phase particles affect the crack growth mechanisms and rates in IN718 alloy. Figure 11b implies the lamellar distribution of eutectic particles in the coating. The cracks growing along the direction of the X–Y axis in this region could be called delamination cracks. Figure 11c is the diagram of the delamination in the IN718 coating. The bending was drawn to make delamination more visible. It can be seen from Table 4 that the content of the element Nb decreases significantly from the enrichment region to the surface. The element Nb plays a role in improving intergranular corrosion resistance. More serious intergranular corrosion occurred above the delamination when the element Nb decreased. In Sanviemvongsak's study [16], intergranular corrosion was observed in the top region of the IN718 coating after high-temperature oxidation. Additionally, it can be found in Figure 10 that the oxygen element diffused from region (I) to region (III), which indicates the intergranular corrosion occurred in the high-temperature oxidation environment. With the decrease in mass gain in Sample 1600-10 after 80 h of high-temperature oxidation treatment, it can be concluded that the material shedding occurred under the combined action of delamination cracks and intergranular corrosion (Figure 11d). In the process of high-temperature oxidation, Nb-rich and second-phase particles were formed, which increased the crack sensitivity of the coating. Hence, the cracks formed in HAZ were more simply extended to the surface of the coating, and consequently, the cracks passed through the thickness of the IN718 coating (Figure 9b).

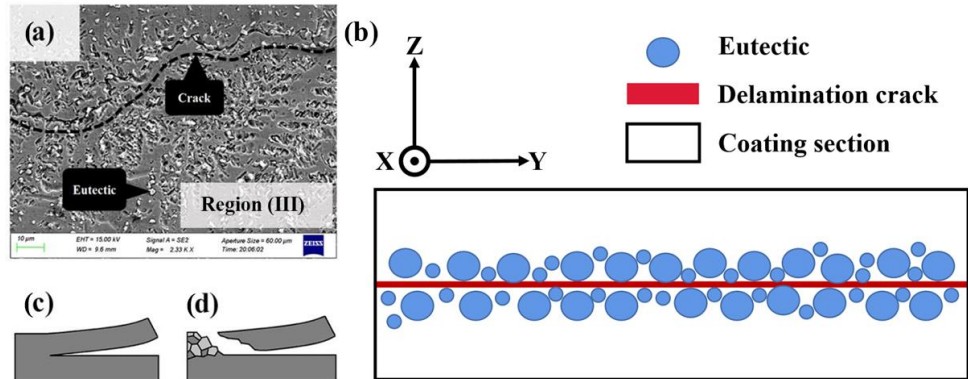

**Figure 11.** Delamination crack below oxide film: (**a**) dense eutectic particles and a crack in region III; (**b**) schematic of lamellar distribution of eutectic particles and a delamination crack; (**c**) delamination in the coating; (**d**) intergranular corrosion and material shedding.

**Table 4.** EDS analysis of Sample 1600-10 after high-temperature oxidation treatment (*wt*%).

| Marked Locations | O | Mo | Cr | Mn | Fe | Ni | Nb |
|---|---|---|---|---|---|---|---|
| I | 6.37 | 8.13 | 7.49 | 0.11 | 28.01 | 48.59 | 1.30 |
| II | 0.86 | 4.71 | 18.83 | 0.37 | 27.50 | 45.51 | 2.22 |
| III | 1.01 | 25.13 | 2.38 | 0.26 | 5.06 | 45.34 | 20.83 |

Figure 12 provides the mass gain variations of the samples with different process parameters under the high-temperature oxidation for 100 h. The mass gain of all samples increased with the increasing treatment time before 80 h. The increase in mass gain contributed to the oxide formation. On the contrary, after 80 h, the mass gain decreased in samples 1600-10 and 1600-15, which indicated the occurrence of material shedding in the IN718 coating, leading to the mass decrease at 100 h.

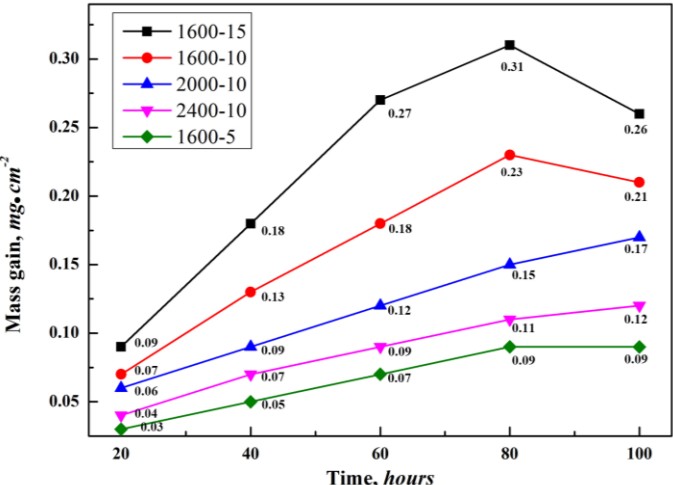

**Figure 12.** Mass gain variations of the samples under high-temperature oxidation for 100 h.

Figure 13 provides the final mass gain of the samples with different process parameters after high-temperature oxidation for 100 h. With the scanning speed increasing from 5 mm/s to 15 mm/s, the mass gain of the sample oxidized at high-temperature increased from 0.09 mg/cm$^2$ to 0.26 mg/cm$^2$, which indicated that the oxidation products of the sample increased, and the high-temperature oxidation resistance decreased. When the laser power increased from 1600 W to 2400 W, the mass gain of the sample decreased from 0.21 mg/cm$^2$ to 0.12 mg/cm$^2$, indicating the improvement of high-temperature

oxidation resistance. The analysis of the oxide film and the microstructural evolution nearby presented a positive correlation between eutectic content and oxide. When the eutectic quantity in the top region of the IN718 coating increased, the oxide on the coating surface also increased after high-temperature oxidation treatment. Combined with the microstructure analysis in the top region (Figure 5,) when the laser power increased, the dendrites in the top region of the coating increased, while the eutectic quantity decreased. Higher laser power reduced the mass gain of the final oxide and enhanced the high-temperature oxidation resistance of IN718 coating. However, with the increase in scanning speed, more eutectic elements precipitated, and more oxides formed in the top region of the coating. In addition, the high-temperature oxidation resistance of the coating decreased.

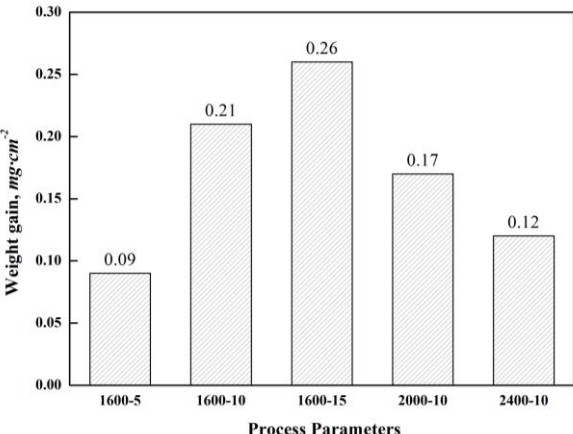

**Figure 13.** Final mass gain of the samples after high-temperature oxidation for 100 *h*.

Two linear regression equations of mass gain and process parameters are shown in Figure 14. When the laser power (*P*) increases by 50%, the mass gain (*M*) decreases by about 43%. When the scanning speed (*V*) is increased by 200%, the mass gain (*M*) is increased by about 188%. The data are also nonlinear-fitted by software (Minitab 2021), and the equation is as follows:

$$\text{Mass gain (mg·cm}^{-2}) = 0.40\, P^{0.0087}\, V^{0.25} - 0.64.$$

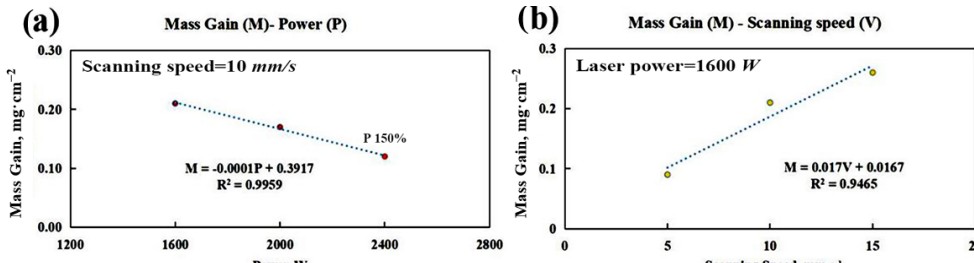

**Figure 14.** Linear regression equation of mass gain and process parameters; power (**a**), and scanning speed (**b**).

This equation is used for the preliminary evaluation of high-temperature oxidation resistance. The scanning speed has a more significant impact on the mass gain of the sample. It can be explained as the effect of crack. In the study of Xing et al. [32], oxides grew on the inner wall of the crack. As a result, the inside of the crack was filled with oxide.

Figure 15 shows the schematic of the high-temperature oxidation mechanism of the surface cracks in the IN718 coating. The inside of the crack was gradually filled with oxide during the high-temperature oxidation treatment (Figure 15a–c). It was considered that when the laser power reduced or the scanning speed increased, the dendrite was refined, and the eutectic compounds increased. Those changes negatively affected the high-temperature oxidation resistance of the IN718 coating. In the range of process pa-

rameters selected in this experiment, with the increase in scanning speed, the coating had higher cracking sensitivity under stress. The crack originated in HAZ in Figure 15d and extended to the surface. The crack extension could increase the surface contact area of the IN718 coating in the oxidation environment and further improve the mass gain of the IN718 coating.

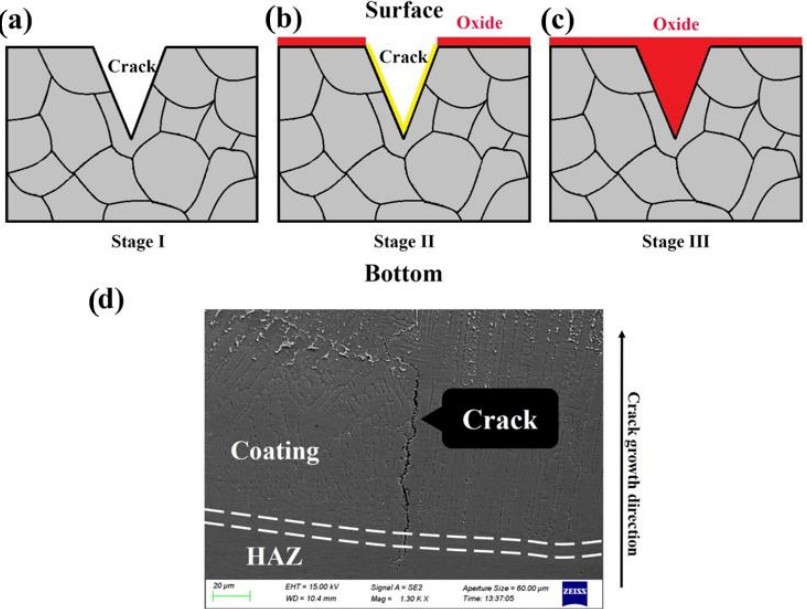

**Figure 15.** Schematic (**a**–**c**) and SEM (**d**) of the high-temperature oxidation mechanism of the surface cracks in the IN718 coating.

The high-temperature behavior of the IN718 coating could be divided into surface oxidation, intergranular corrosion, and material shedding. The purpose of changing process parameters was to adjust the precipitation of eutectic elements and reduce the mass gain following high-temperature oxidation. Cracks were the leading cause of coating failure. The microstructure evolution of coating and substrate should be considered comprehensively in the preparation of high-temperature coating to prevent cracks in the coating and HAZ. Figure 16 shows the relationship between process parameters, microstructure, and high-temperature oxidation resistance, where the positive sign (+) indicates an increase, while the negative (−) sign indicates a decrease.

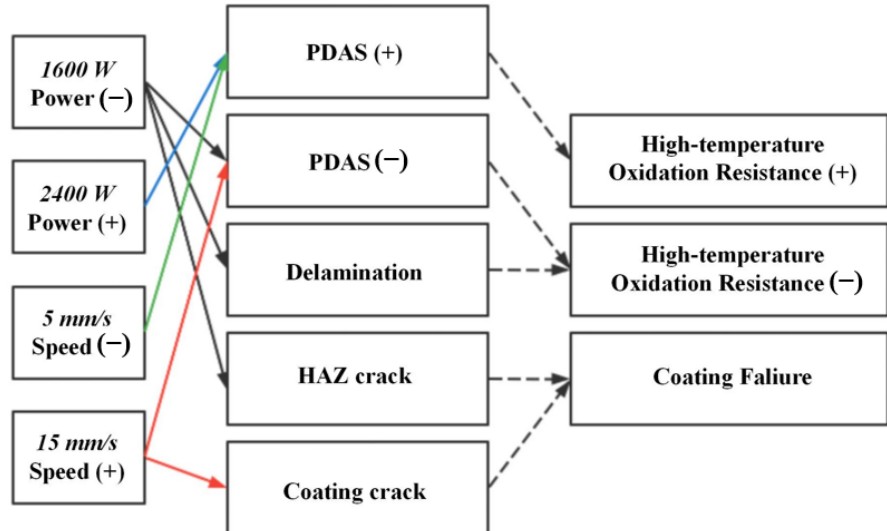

**Figure 16.** Effects of process parameter adjustment on high-temperature oxidation resistance.

### 3.5. Engineering Application

Figure 17 demonstrates a remanufacturing process of a Cr5Mo hydrocracking furnace tube for a petrochemical enterprise. The design life of the Cr5Mo furnace tubes was more than 15 years, and the actual service time was 15 years. In Figure 17a, severe oxidation has occurred on the surface of the Cr5Mo furnace tube. In Figure 17b, the surface of the furnace tube is processed to remove the oxide scale. After turning to remove the surface oxide layer and defects, the thickness of the original Cr5Mo tube complied with the standard ISO 13704. Penetrant testing was used to find surface defects, and the surface roughness value was Ra 6.3 (profile arithmetic average error, μm). Laser cladding IN718 coating on the surface of the Cr5Mo furnace tube is shown in Figure 17c. In Figure 17d, the surface of the furnace tube after laser cladding is processed to be smooth, resizing according to the requirements of engineering drawings. In the process of laser cladding, in regard to the experimental results, a selected laser power of 2400 W and scanning speed of 10 mm/s prepared the IN718 coating without significant defects after ultrasonic non-destructive testing and surface penetrant testing. The desired laser power of 2400 W can prevent the grain boundary liquefaction at low laser power. Scanning speed has a more significant impact on the high-temperature oxidation resistance of IN718 coating. Hence, the selected scanning speed of 10 mm/s can reduce the crack sensitivity of HAZ. Afterwards, this paper excluded the scanning speed of 5 mm/s for the purpose of ensuring the laser cladding efficiency. According to the experimental results, the Sample 2400-10 exhibited favorable high-temperature oxidation resistance. When the power was 2400 W and the scanning speed was 10 mm/s, there were no defects in FZ and HAZ. The laser cladding process could be divided into several stages to prevent the thermal deformation of the tube in excessive heating. The temperature of the Cr5Mo substrate during the laser cladding process ought to be strictly controlled below 600 °C.

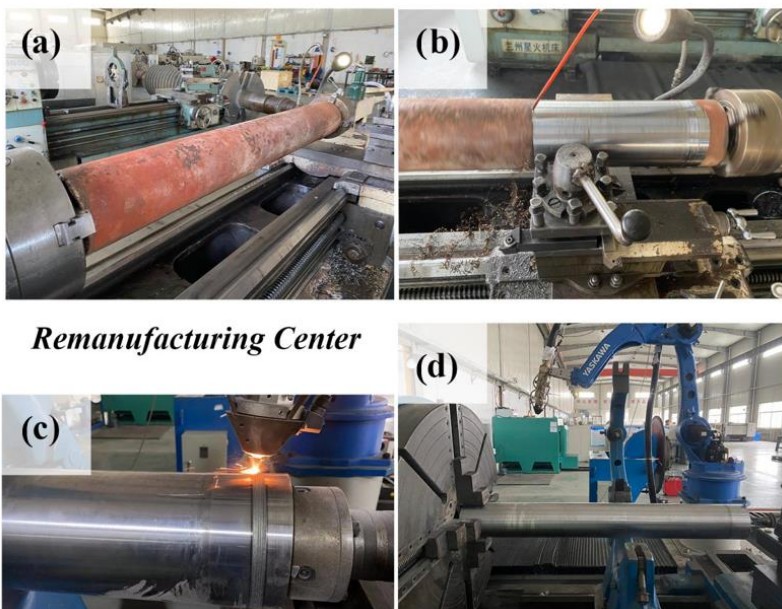

**Figure 17.** Remanufacturing of Cr5Mo hydrocracking furnace tube for a petrochemical enterprise: (**a**) serious oxidation of the Cr5Mo furnace tube; (**b**) surface processing of the furnace tube to removed oxide scale; (**c**) laser cladded IN718 on the surface of the Cr5Mo furnace tube; (**d**) surface processing and resizing after laser cladding.

The Larson–Miller equation LMP = T1 [C + log (t)] can be used to evaluate the remaining life of the tubes [33], where LMP is the Larson–Miller parameter, T1 is the temperature (in Kelvin), and t is the time to rupture. The value of the C-parameter locates between 15 and 30. It can be seen from the equation that the remaining life of the furnace tubes is significantly affected when overtemperature occurs. The high-temperature properties of

materials play a vital role in the service life of furnace tubes. In the high-temperature creep test, when the Cr5Mo is performed at 550 °C and 190 MPa [34] and IN718 is performed at 700 °C and 560 MPa [35], both have the same stress rupture time of approximately 70 h. As IN718 has higher stability above 600 °C, it is helpful to improve the service life of remanufactured Cr5Mo furnace tubes in high-temperature oxidation environments. Figure 18 records the overhaul period of the Cr5Mo furnace tube in a certain petrochemical enterprise since 2019, which partly proves this conclusion. Deep cracks were found on the original Cr5Mo furnace tube without IN718 protective coating in the last testing. Thus, the remanufactured Cr5Mo furnace tube has longer service life than the original Cr5Mo furnace tube under the same operating conditions.

| Type of furnace tube | Temperature | Examine and repair period | | | |
|---|---|---|---|---|---|
| | | 2019.10 | 2020.10 | 2021.10 | 2022.10 |
| (A) Original Cr5Mo | 500°C –700°C | Normal | Normal | Normal | Failure |
| (B) Remanufactured Cr5Mo | | Normal | Normal | Normal | Normal |

**Figure 18.** Overhaul period of the (**A**) original and (**B**) remanufactured Cr5Mo furnace tube in a certain petrochemical enterprise.

## 4. Conclusions

This study revealed the relationship between microstructure and high-temperature oxidation resistance in laser cladding IN718 coating, which proposed a loss mechanism of the coating under the combined action of delamination cracks and high-temperature corrosion. The experimental results were applied in a practical application. However, the application requires feedback after sufficient running time because various factors occur in the service environment. The main conclusions of the current study can be summarized as follows:

1. When the scanning speed increased from 5 mm/s to 15 mm/s, the top region's dendrites refined, and eutectic quantity increased. When the laser power increased from 1600 W to 2400 W, the dendrite in the top region coarsened, and the eutectic quantity decreased. Lower power and faster scanning speed improve the cooling rate, resulting in increased precipitation of eutectic elements. The change of laser process parameters affected the number of strengthening phases precipitated in the parent phase and subsequently changed the hardness of the coating.

2. When the scanning speed was increased from 5 mm/s to 15 mm/s, the average hardness of HAZ was increased by about 26 %. Under the condition of laser heating and rapid cooling, martensite with high hardness was generated in the HAZ region. Nevertheless, the brittleness of martensite also increased the crack sensitivity of HAZ, and the thermal fatigue cracks in HAZ initiated under thermal cyclic loading. The HAZ cracks along the thickness of the coating, resulting in coating failure.

3. When the scanning speed was increased from 5 mm/s to 15 mm/s, the mass gain was increased by about 188%. When the laser power increased from 1600 W to 2400 W, the mass gain decreased by about 43%. The formation of oxide was closely related to the distribution of the Ni3Nb phase in the top region of the coating. The less eutectic precipitation, the better the high-temperature oxidation resistance of IN718 coating. The scanning speed had a more significant impact on the high-temperature oxidation resistance.

4. The high-temperature behavior of the IN718 coating can be divided into surface oxidation, intergranular corrosion, and material shedding. The combined action of delamination cracks and intergranular corrosion led to material shedding. Crack

extension increased the surface contact area in the oxidation environment, which negatively affected the high-temperature oxidation resistance of IN718 coating.

**Author Contributions:** Conceptualization, Z.X. and F.W.; methodology, Z.X.; software, Z.X. and J.G.; validation, Z.X., W.L., and S.P.; formal analysis, Z.X.; investigation, Z.X.; resources, F.W. and W.L.; data curation, Z.X.; writing—original draft preparation, Z.X.; writing—review and editing, F.W. and S.P.; visualization, J.G.; supervision, F.W.; project administration, F.W.; funding acquisition, F.W. All authors have read and agreed to the published version of the manuscript.

**Funding:** This work was supported by the National Natural Science Foundation of China (Grant No. 51875075), the Natural Science Foundation of Guangdong Province (Grant No. 2021A1515011989), the Guangdong Provincial University Innovation Team Project (Grant No. 2020KCXTD012), the 2020 Li Ka Shing Foundation Cross-Disciplinary Research (Grant No. 2020LKSFG01D), the Key Projects of Universities in Guangdong Province (Grant No. 2019KZDXM046), and Shantou University Research Startup Funding Project (Grant No. NTF19030), the Equipment Pre-research Foundation (No. 80923010401).

**Institutional Review Board Statement:** Not applicable.

**Informed Consent Statement:** Not applicable.

**Data Availability Statement:** Not applicable.

**Conflicts of Interest:** The authors declare no conflict of interests.

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
