# Peer review of "Effects of Process Parameters on Microstructure and High-Temperature Oxidation Resistance of Laser-Clad IN718 Coating on Cr5Mo Steel"

_coatings, doi:10.3390/coatings13010197_

Round 1

Reviewer 1 Report

Please see the enclosed comment file.

Reviewer 2 Report

In line 318 the effect of the process parameters is compared by evaluating the inclination of the linear regression line. It would be advisable to compare the effects on the outputs with the amplitude of the range of variation of the 2 input parameters (50% for laser power and 200% for scanning speed).

Reviewer 3 Report

The Authors' manuscript fits perfectly into the technical-scientific profile of the journal "Coatings". The topics discussed, in general (damage, surface treatment, lifetime management / enhancement), have topicality. The equipment and methods used by the Authors are state-of-the-art, but do not make up for the low number of samples. Throughout the reading of the manuscript, there is a sense that the relationship between the aims and the tools is not in the right place. (In order to achieve the aims, the Authors use the tools, or the possibilities given by the tools determine the aims.)

To describe the experiments (Figure 2), the geometric dimensions of the pipe (substrate), the distance between the coating layers and the height of the coating must be specified. After the first coating is applied, to what temperature has the tube cooled before the second coating is applied (and so on, as appropriate)? Could the thermal effects on the pipe (substrate) have had an effect on the results?

Can the Authors please provide the hardness values in Figure 7 accurately? What is the relationship between the data in Table 3 and the data in Figure 7?

Detail b) of Figure 9 shows fatigue cracking, according to the Authors. Why is fatigue cracking found in the sample? What is the source of cyclic load / stress in our case? What is the relationship between detail b) in Figure 9 and detail d) in Figure 15? The data in Figure 9 are not legible and should be made visible.

Figure 12 b) is not sufficiently illustrative; the spatial representation does not help but hinders interpretation.

I would like to ask Authors to explain the following sentence (rows 274-275): "The bending was drawn to make delamination more visible."

For details of Figure 14, please add the relevant scanning speed and laser power values. The equation in line 321 is not understandable in its present form, please clarify.

I think that Figure 16 is for summary purposes. Was the number of samples sufficient to produce such a figure with a generalising purpose? (Less is sometimes more.)

Authors use the Larson-Miller method in Section 3.5 of the manuscript. Are the conditions for using this method met?

I suggest Authors to consider the contents of the "Conclusions" section.

Redundant elements of the manuscript are suggested to be removed: rows 44-47; rows 62-71; rows 173-175 and 196-197. I do not consider the wording "Cr5Mo tubular furnace" as it is a steel material for the furnace to be correct. Please correct typing errors, e.g. row 31, rows 212-214, row 406. I suggest considering the use of subscripts in the symbols for compounds.

Round 2

Reviewer 2 Report

The paper “Effects of process parameters on microstructure and high-temperature oxidation resistance of laser cladding IN718 coating on Cr5Mo steel” explored the effect of process parameters (scanning speed and laser power) on the high-temperature oxidation resistance of laser cladding IN718 coating on Cr5Mo tube by means.

The authors comprehensively responded to the points presented in the previous review round. The manuscript has been modified in accordance with what was previously reported and the bibliography has been expanded and made exhaustive.  I suggest publishing the work in the present form

Author Response

Thank you for your affirmation.

Reviewer 3 Report

Thanks to the Authors for the additions and modifications highlighted in green in the manuscript and included in the "Response to Reviewer 3 Comments".

I can accept the Authors' response that “this study can explain the influence mechanism of process parameters on high temperature oxidation resistance, and guide the actual production”.

I thank Authors for their additions regarding the geometric dimensions of the tube (substrate) and coating. Authors have answered my question regarding the temperatures measured and achieved during the preparation of the coating, I accept the answers. However, the answer is not included in the revised manuscript, only in the response file. I consider it necessary to add this information to the manuscript. Authors have also answered my question about the hardness values in Table 3 and Figure 7. The answer is accepted, but I consider it necessary to include its content in the manuscript. I thank you for the addition concerning fatigue cracking and for the modification of the relevant text; I believe that these were substantive modifications. Unfortunately, the details in Figure 9 are still not legible and should be made visible. I still maintain that the detail in Figure 12(b) is not sufficiently illustrative; the spatial representation does not help but hinders interpretation. Authors have explained the content of the sentence “[t]he bending was drawn to make delamination more visible”, please add the explanation to the manuscript. Authors have described in their response the additions required for Figure 14; please include these in the figure. The formula in line 331 (new numbering) is now clear in this form, but why it is not on a separate line is not clear. Please consider this. I accept the Authors' response to Figure 16. I also accept the addendum on the applicability of the Larson-Miller method. The contents of the "Conclusions" section in the revised manuscript can be described as relatively straightforward. Again, I ask Authors to consider the content of the section (explanations / interpretations are missing).

Thanks for corrections related to redundancies, typing errors and subscripts.

Round 3

Reviewer 3 Report

Thanks to the Authors for the additions and modifications highlighted in green in the manuscript and included in the document titled "Response to Reviewer 3 Comments (2)".

Authors have included in the manuscript their answers to questions that were previously only contained in the "Response to Reviewer 3 Comments". I have previously accepted these responses and thank you for publishing them. Figures 7, 9, 12 and 14 and their corresponding text have been modified; the clarity of the figures has been improved. The formula in the text has been highlighted. An additional point has been added to the "Conclusions" section and two other points have been changed by the Authors.

I recommend publication of the revised manuscript.